# Adverse impact of female reproductive signaling on age-dependent neurodegeneration after mild head trauma in *Drosophila*

Changtian Ye[1], Ryan Ho[2], Kenneth H Moberg[1], James Q Zheng[1,3,4]*

[1]Department of Cell Biology, Emory University School of Medicine, Atlanta, United States; [2]College of Art and Science, Emory University, Atlanta, United States; [3]Department of Neurology, Emory University School of Medicine, Atlanta, United States; [4]Center for Neurodegenerative Diseases, Emory University School of Medicine, Atlanta, United States

**\*For correspondence:**
james.zheng@emory.edu

**Competing interest:** The authors declare that no competing interests exist.

**Abstract** Environmental insults, including mild head trauma, significantly increase the risk of neurodegeneration. However, it remains challenging to establish a causative connection between early-life exposure to mild head trauma and late-life emergence of neurodegenerative deficits, nor do we know how sex and age compound the outcome. Using a *Drosophila* model, we demonstrate that exposure to mild head trauma causes neurodegenerative conditions that emerge late in life and disproportionately affect females. Increasing age-at-injury further exacerbates this effect in a sexually dimorphic manner. We further identify sex peptide signaling as a key factor in female susceptibility to post-injury brain deficits. RNA sequencing highlights a reduction in innate immune defense transcripts specifically in mated females during late life. Our findings establish a causal relationship between early head trauma and late-life neurodegeneration, emphasizing sex differences in injury response and the impact of age-at-injury. Finally, our findings reveal that reproductive signaling adversely impacts female response to mild head insults and elevates vulnerability to late-life neurodegeneration.

## eLife assessment

The authors have presented an interesting set of results showing that female sex peptide signaling adversely affects late-life neurodegeneration after early-life exposure to repetitive mild head injury in *Drosophila*. This **fundamental** work substantially advances our understanding of how sex-dependent response to TBI occurs by identifying the sex peptide and the immune system as modulators of sex differences. The evidence supporting the conclusions is **compelling** with rigorous inclusion of controls and appropriate statistics.

## Introduction

Neurodegenerative disorders affect millions of people worldwide, and this number is expected to increase substantially with rising life expectancy and population growth. Major efforts have been directed toward understanding the etiology of neurodegenerative conditions, but the challenge remains to elucidate molecular mechanisms underlying the onset and progression of neurodegeneration. Recent studies have identified multiple genetic mutations and risk factors associated with dementia and neurodegeneration, but additional causes are likely involved (*Fu and Ip, 2023*;

*Pihlstrøm et al., 2017*; *Bertram and Tanzi, 2005*; *Hardy and Orr, 2006*). Aging is a major risk factor for many neurodegenerative disorders and plays a key role in disease emergence and progression (*Azam et al., 2021*; *Saikumar and Bonini, 2021*). Other significant risk factors include environmental insults that can evoke neuropathological processes, which may be latent at first but surface with aging to disrupt various aspects of brain structure and function, leading to neurodegenerative conditions (*Brown et al., 2005*; *Gupta and Sen, 2016*). Physical blows to the head are known to cause symptoms related to immediate brain injury, but even seemingly innocuous head impacts are strongly linked to brain dysfunction and degeneration later in life (*Gupta and Sen, 2016*; *Brett et al., 2022*; *Ntikas et al., 2022*). It is thus crucial to understand the aberrant processes triggered by these mild physical insults that may be initially 'hidden' but can give rise to later neurodegenerative conditions. Furthermore, sex differences have been documented in many degenerative diseases, including chronic brain degeneration following mild traumatic brain injury (*Yue et al., 2019*; *Berz et al., 2013*; *Gupte et al., 2019*). How sex and age contribute to an individual's response to mild brain disturbances and modify the risk for neurodegeneration remains to be fully understood.

The fruit fly, *Drosophila melanogaster*, represents an excellent and tractable model organism for dissecting fundamental disease mechanisms, including neurodegeneration (*Hirth, 2010*; *Lessing and Bonini, 2009*; *Petersen and Wassarman, 2012*; *McGurk et al., 2015*). Importantly, the fruit fly's relatively short lifespan enables longitudinal interrogation of disease progression from the initial trigger to the late-life emergence of neurodegenerative conditions such as Alzheimer's disease, amyotrophic lateral sclerosis, and frontotemporal dementia (*McGurk et al., 2015*; *Pandey and Nichols, 2011*; *Yuva-Aydemir et al., 2018*; *Bolus et al., 2020*; *Prüßing et al., 2013*). Recently, several *Drosophila* head injury models have been developed that recapitulate key findings from traumatic brain injury (TBI) in humans and other preclinical models, revealing significant insights into the potential molecular and genetic underpinnings of injury responses (*Barekat et al., 2016*; *Katzenberger et al., 2013*; *Sun and Chen, 2017*; *Saikumar et al., 2020*; *van Alphen et al., 2022*; *Ye et al., 2023*; *Behnke et al., 2021b*). We recently developed a novel *Drosophila* model (HIFLI: Headfirst Impact FLy Injury) in which mild repetitive head-specific impacts can be delivered to multiple awake and unrestricted adult flies of both sexes (*Behnke et al., 2021b*). In this study, we utilized the HIFLI model to deliver a milder version of repetitive head trauma, which eliminates potentially confounding effects from injury-induced death and enables us to interrogate sex and age-at-injury contributions to the emergence of brain deficits throughout the entire lifespan. Our data show that exposure to this milder form of head trauma elicits minimal acute deficits but causes profound brain-associated behavioral deficits and brain degeneration that only emerge later in life. These late-life neurodegenerative conditions are further exacerbated by increasing age-at-injury and disproportionately elevated in mated females. We further identify that sex peptide signaling involved in female reproduction plays a key role in elevated female neurodegeneration after mild head trauma. Finally, RNA sequencing data suggest that the chronic suppression of innate immune defense networks in mated females may mediate the elevated vulnerability to neurodegeneration after mild head trauma. Together, our results further establish *Drosophila* as an excellent model system to investigate lifelong neurodegenerative conditions, highlight that sex and age significantly contribute to trauma recovery and outcome, and support the notion that physical insults to the head, even very mild ones, pose a major threat for brain health.

## Results

### Exposure to a milder form of head impacts elicited no mortality but resulted in age-related acute sensorimotor deficits

The HIFLI system delivers head-first impacts to multiple adult flies at ~5 m/s (*Behnke et al., 2021b*). Our previous study employing the HIFLI model delivered two separate sessions of 15 iterative impacts, with a 1-day interval between the two sessions. Young flies exposed to this injury paradigm exhibited a small percentage of acute mortality and a significant reduction in lifespan, but flies subjected to one session exhibited little to no acute mortality (*Behnke et al., 2021b*). To enable lifelong investigation of the age-at-injury as a risk factor for neurodegeneration with minimal deaths, we reduced the severity of our paradigm to only one session of 15 impacts, hereafter referred to as very mild head trauma (vmHT). We then quantified survival, sensorimotor deficits, and brain pathology at various timepoints after exposure to vmHT (*Figure 1A*). As reported in our previous publication, around 50% of flies

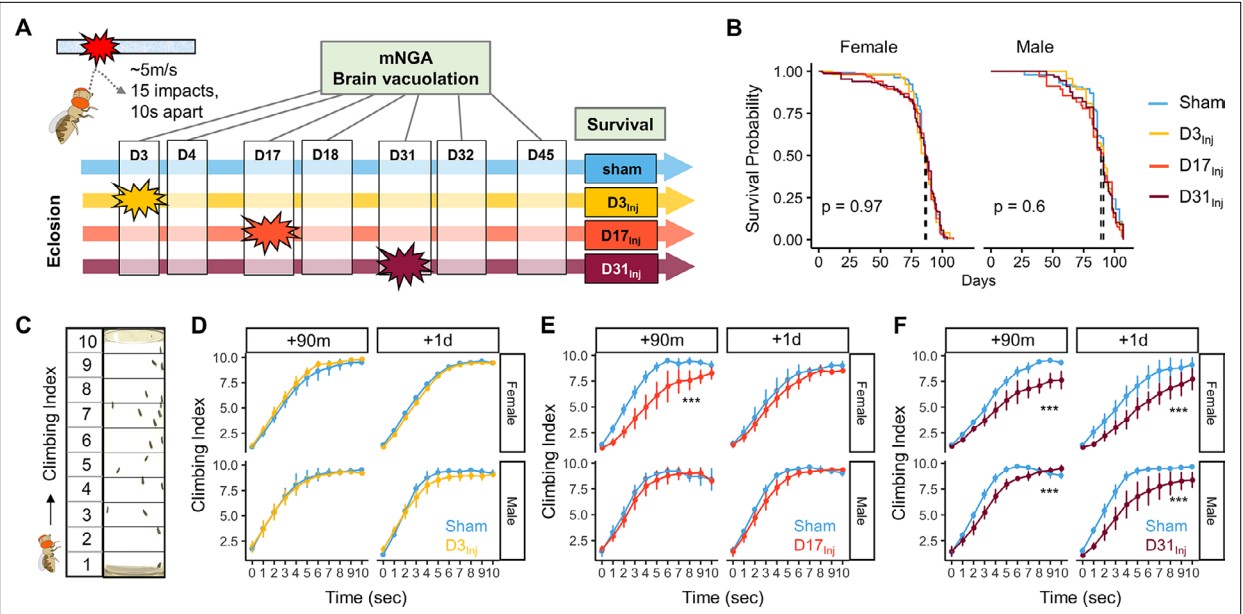

**Figure 1.** Exposure of adult fruit flies to a very mild form of repetitive head impacts at different ages elicits minimal acute effects. (**A**) Schematic illustration of the experimental design. Male and female flies were exposed to very mild head trauma (vmHT) on D3, D17, or D31 post-eclosion (denoted as D3$_{Inj}$, D17$_{Inj}$, and D31$_{Inj}$), whereas the sham groups did not receive vmHT. Sensorimotor behavior and brain pathology were assessed at 90 min and 1 day after vmHT exposure using modified negative geotaxis assay (mNGA) and imaging-based quantification of vacuolization. (**B**) Survival curves showing no significant alteration in lifespan among different age-at-injury groups and between sexes. Kaplan–Meier p-values were determined using the Mantel–Cox log rank test with Bonferroni correction. Total N = 871, n > 50 flies in each condition. (**C**) Schematic diagram depicting mNGA where the Climbing Index (CI) at each second of the 10 s trial was calculated using the number of flies in each of the 10 height bins (see 'Materials and methods' for details). (**D–F**) CI plots depicting acute effects of vmHT on the climbing behavior of D3$_{Inj}$, D17$_{Inj}$, and D31$_{Inj}$ cohorts. (**D**) Both male and female D3$_{Inj}$ flies showed no deficits in climbing at 90 min and 1 day post-injury. (**E**) Only female D17$_{Inj}$ flies exhibited climbing deficits at 90 min post-injury (***p=5.75e-08) but they recovered 1 day post-injury. (**F**) Both male and female D31$_{Inj}$ flies exhibited marked decline in climbing ability at both time points (p-values: 5.36e-08 and 6.2e-06 for female + 90m and +1d; 0.00033 and 1.91e-07 for male +90m and +1d). A total of nine videos were used for each condition: three experimental repeats of three trials, number of flies in each video n ≥ 10. Error bars: standard error (±se). Repeated-measures ANOVAs were conducted to examine the effects of injury on climbing indices at each second.

exhibit concussive-like behaviors immediately after receiving headfirst impacts, including uncoordinated movements, freezing, and loss of postural stability, but all flies recovered fully within the first 5 min (*Behnke et al., 2021b*). Importantly, vmHT had no significant effects on both short- and long-term survival when delivered on post-eclosion day 3, 17, or 31 (*Figure 1B*), confirming the very mild nature of our injury paradigm.

Sensorimotor behaviors were assessed using a modified negative geotaxis assay (mNGA, see 'Materials and methods'). We performed mNGA at 90 min and 1 day post-injury (1 dpi) following vmHT exposure to examine acute injury effects and potential short-term recovery. Using fly positional data, we calculated a Climbing Index (CI) for each second of the 10 s trial (*Figure 1C*). Male and female flies injured on day 3 post eclosion (D3$_{Inj}$) did not exhibit any acute deficits in climbing behavior (*Figure 1D*), though this was not the case for flies injured on day 17 (D17$_{Inj}$) or day 31 (D31$_{Inj}$). We noted a significant reduction in CI in D17$_{Inj}$ females at 90 min post-injury, but not in males of the same age cohort (*Figure 1E*). These females appeared to have mostly recovered by 24 hr post-injury (*Figure 1E*). In comparison, both male and female D31$_{Inj}$ cohorts exhibited a substantial reduction in CI at 90 min as well as 1 dpi (*Figure 1F*), indicating age-related acute sensorimotor deficits after vmHT exposure and impaired recovery. These results are consistent with the notion that aging worsens post-head injury recovery (*Katzenberger et al., 2016*) and that older populations are at a higher risk of developing negative consequences following mild head injury (*Abdulle et al., 2018*; *Peterson et al., 2014*; *Mosenthal et al., 2004*).

We next examined the short-term effects of vmHT on brain structure using a previously described protocol to stain whole fly brains and to detect vacuoles (*Behnke et al., 2021a*). Vacuolation, as seen in *Drosophila* models of neurodegenerative diseases, is a marker for frank tissue degeneration in the

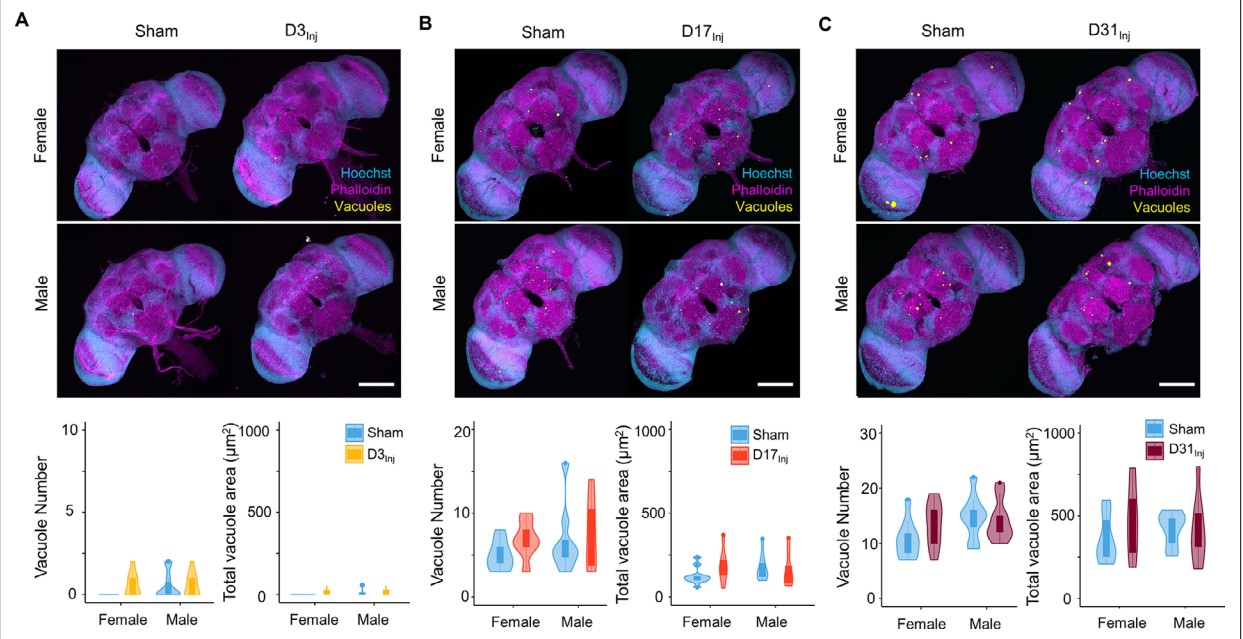

**Figure 2.** Exposure to very mild head trauma (vmHT) does not acutely increase vacuole formation in the brain. (**A–C**) Two-photon whole brain imaging and quantification of vacuoles in D3$_{Inj}$ (**A**), D17$_{Inj}$ (**B**), and D31$_{Inj}$ (**C**) flies. Top panels: representative z-projected whole-brain images of different injury groups with their respective sham brains of both sexes. Vacuoles are highlighted by yellow color. Scale bar = 100 μm. Bottom: violin plots and boxplots represent quantification of vacuole number and total vacuole area in each condition. Boxplots whiskers correspond to the maximum 1.5 interquartile range. Two experimental replicates resulting in n > 10 brains for all conditions. Stats: nonparametric Wilcoxon rank-sum tests, p>0.05.

The online version of this article includes the following figure supplement(s) for figure 2:

**Figure supplement 1.** Exposure to very mild head trauma (vmHT) on D3 does not elicit gross morphological alterations in pigment dispersing factor (PDF) neurons and mushroom bodies.

**Figure supplement 2.** Exposure to very mild head trauma (vmHT) does not elicit significant apoptosis.

fly brain (*Heisenberg and Böhl, 1979*; *Sunderhaus and Kretzschmar, 2016*). These roughly round lesions can range from 1 to 50 μm in diameter and are largely located within the neuropil. Though aging alone can increase the number and size of vacuoles in the brain, we demonstrate that vmHT did not generate any changes in vacuolation at 1 dpi in all injury groups (*Figure 2A–C*). Examination of the mushroom body of the central nervous system and the pigment dispersing factor (PDF) neurons in the optical lobe of D3$_{Inj}$ flies also found no obvious structural alterations shortly after vmHT exposure (*Figure 2—figure supplement 1*). Therefore, the observed acute climbing deficits were unlikely caused by acute structural disruptions of the brain after vmHT. Additionally, we performed TUNEL staining at 1 dpi to investigate whether vmHT causes apoptosis within the brain. As a positive control, we subjected a group of flies to an eye-stabbing procedure aimed at producing a very severe injury capable of inducing apoptotic cell death within the brain (*Sanuki, 2020*). We found that brains from the eye-stabbing injury group clearly displayed TUNEL-positive signals near primary injury regions, validating our TUNEL stain (*Figure 2—figure supplement 2*). However, TUNEL staining was faint in the brains of both the shams and vmHT-exposed groups. Consistently, we did not observe any exoskeletal or eye damage immediately following injuries, nor did we observe any retinal degeneration and pseudopupil loss at the chronic stage of these flies. These data further support the mild nature of our head trauma model and suggest that acute sensorimotor deficits were not a direct result of neuronal death nor caused by gross brain disruptions.

## Exposure to vmHT caused delayed onset of neurodegenerative conditions

Physical insults to the head, even very mild cases, represent a risk factor for the development of neurodegenerative conditions later in life. While exposure to vmHT did not generate obvious short-term brain-associated deficits in our younger cohorts (D3$_{Inj}$ and D17$_{Inj}$), we hypothesized that they

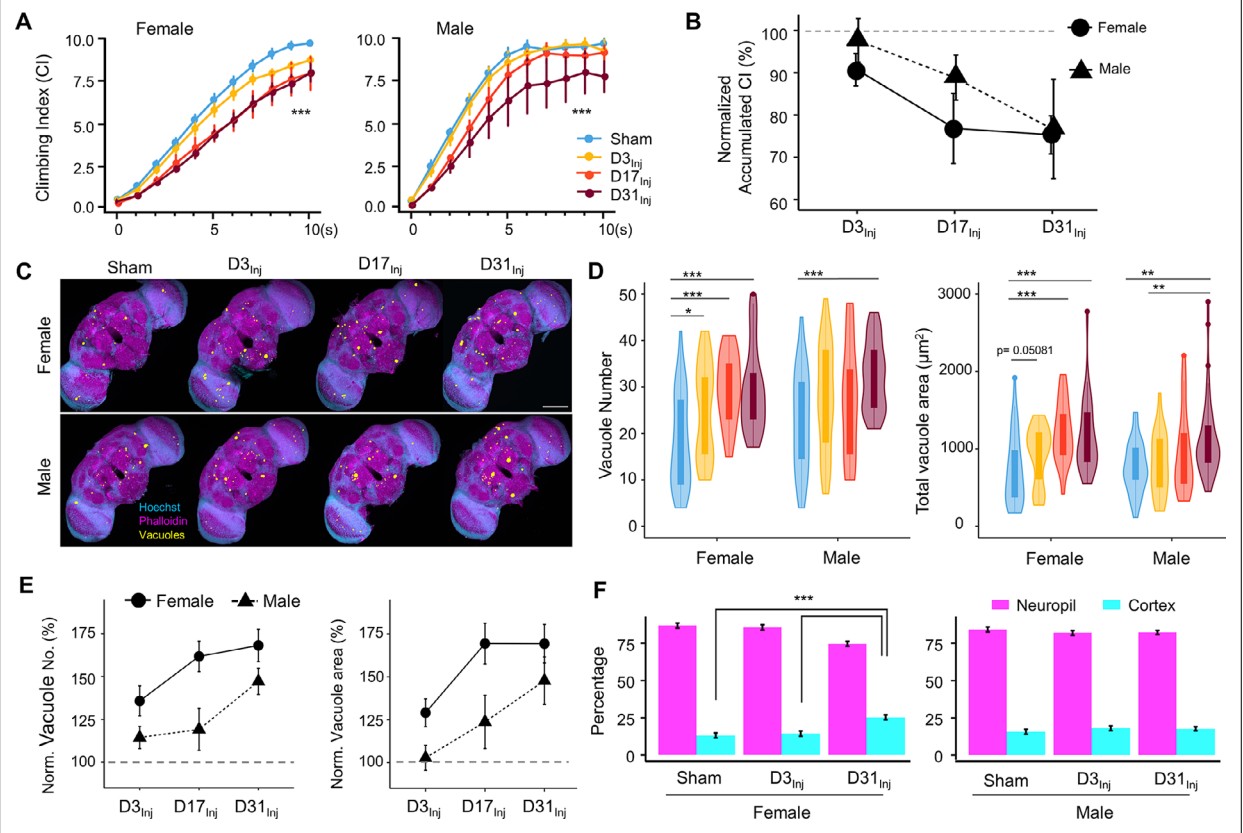

**Figure 3.** Exposure to very mild head trauma (vmHT) results in late-life brain deficits and neurodegeneration. (**A**) Exposure to vmHT at various ages altered negative geotaxis behavior when assessed on D45. A total of nine videos were used for each condition: three experimental repeats of three trials, number of flies in each trial n ≥ 10. Error bar: ± se. Repeated-measures ANOVA and Bonferroni post hoc tests were conducted to examine the effects of injury on climbing indices at each second. Overall, vmHT exposure reduced Climbing Index (CI) for females (***p=2e-16) and males (***p=1.2e-11). See ***Supplementary file 1c*** for p-values from pairwise comparisons by injury conditions and time. (**B**) Sex differences in vmHT-induced climbing impairment are associated with age-at-injury. D3$_{Inj}$ and D17$_{Inj}$ females exhibit a stronger decline in normalized accumulated CI compared to males. D31$_{Inj}$ females and males, on the other hand, suffered a similar substantial reduction in accumulated CI. Accumulated CI data of injury groups are normalized to their respective sham levels. (**C**) Representative z-projected whole images depicting vacuole formation in each condition (sex/injury group). Scale bar = 100 μm. (**D**) Quantification of vacuole number and total vacuole area in each condition (total N = 352, n > 30 in each condition). Here all sham groups were combined by sex. Boxplots whiskers correspond to the maximum 1.5 interquartile range. Statistics: nonparametric Wilcoxon rank-sum tests. See ***Supplementary file 1c*** for p-values from pairwise comparisons by injury conditions. (**E**) Sex differences in the increase of brain vacuolation were partially dependent on age-at-injury. Vacuole number and total vacuole area of each injury condition were normalized to their respective sham groups. Overall, females exhibited a higher percentage of increase in vacuolation than males. (**F**) Quantification of the percentage of vacuoles in neuropil and cortex region of the brain in sham, D3$_{Inj}$, and D31$_{Inj}$ groups of both sexes. Data was from two independent trials of n = 15 brains in each condition. Two-sample *t*-tests were used to compare the percentage of vacuoles in neuropil and cortical region between each injury condition. D31$_{Inj}$ females had significantly less percentage of vacuoles in the neuropil region compared to the sham group (***p=2.19493e-6).

The online version of this article includes the following figure supplement(s) for figure 3:

**Figure supplement 1.** Age-at-injury affects post-injury recovery after 2 weeks.

could develop neurodegenerative conditions late in life. We quantified sensorimotor behavior and brain vacuolation for sham, D3$_{Inj}$, D17$_{Inj}$, and D31$_{Inj}$ cohorts on day 45 post-eclosion (D45). In this age-matched comparison, we found that vmHT exposure decreased CI regardless of age-at-injury, though injury at older ages induced greater sensorimotor decline (***Figure 3A***). Importantly, sensorimotor deficits were sexually dimorphic as female flies appeared to perform worse than males in the same injury condition. To better visualize the sex differences, we plotted the accumulated CI (summation of CI values over the 10 s trial) and normalized the value to that of the respective sham groups (***Figure 3B***). In the D3$_{Inj}$ cohort, only female flies exhibited significant climbing deficits compared to the sham, whereas D17$_{Inj}$ cohort females had a stronger decline in climbing ability than the males. Here, male flies seemed to be more resilient to vmHT when exposed at younger ages. When injured on D31,

however, both male and female flies exhibited similarly profound sensorimotor deficits when assessed on D45.

For female flies, significant increases in vacuole number and total area were observed for all three age-at-injury cohorts on D45 (*Figure 3C and D*). For male flies, only the oldest injury cohort (D31$_{Inj}$) showed a significant increase in vacuolation. When vacuole number and area were normalized to the respective sham groups and compared between sexes, we found that female brains exhibited a higher percentage of injury-induced increase in vacuolation overall, but particularly in younger injury cohorts D3$_{Inj}$ and D17$_{Inj}$ (*Figure 3E*).

In congruence with our previous finding that aging increases vacuolation, we found that flies aged to D45 exhibited a higher extent of brain vacuolation compared to younger flies, even without injury. To further probe the origin of brain vacuolation, we examined the distribution of vacuoles by quantifying the percentage of vacuoles within the cell-body-rich cortex and the axon-rich neuropil for sham, D3$_{Inj}$, and D31$_{Inj}$ flies on D45. We confirmed that regardless of injury condition, most vacuoles can be found in the neuropil and not in the cortex (*Figure 3F*). Interestingly, D31$_{Inj}$ females have a higher percentage of cortical vacuoles compared to sham and other injury cohorts, but males have a consistent neuropil-to-cortex ratio in all cohorts. Overall, this finding further suggests that neuronal death is not the major contributor of vacuole formation. Indeed, TUNEL staining at this time point (D45) did not yield significant signals for sham, D3$_{Inj}$, D17$_{Inj}$, or D31$_{Inj}$ flies (see *Figure 2—figure supplement 2D*).

The above age-matched comparison highlighted that though vmHT exposure early in life has minimal acute effects, it can result in neurodegenerative conditions in advanced ages. While it inherently makes sense to compare flies at the same age (D45), different injury groups technically experienced different post-injury recovery times in such a comparison. Therefore, we also compared the groups by matching their recovery time of 2 weeks. We found that within the same timeframe, flies injured at younger ages exhibit a smaller decrease in sensorimotor abilities (*Figure 3—figure supplement 1A and B*). This was true for both male and female flies, although sex difference existed in this 2-week comparison as well. Out of all injury groups, D3$_{Inj}$ males were the only cohort of flies that displayed no change in climbing ability 2 weeks after vmHT exposure. On the other hand, all injured females, as well as D31$_{Inj}$ males, exhibited an increase in vacuole formation (*Figure 3—figure supplement 1C–F*). Overall, this recovery time-matched data suggests that aging accelerates the development of neurodegenerative conditions.

## AI-assisted tracking revealed disruptions in fly speed and directional movement

The implementation of the CI in mNGA allows us to assess group climbing behavior and injury-induced deficits with an increased resolution in height (10 height bins) and time (every second). However, mNGA lacks the ability to reveal detailed information regarding sensorimotor behavior of individual flies, such as instantaneous speed and directional angle. Here, we took advantage of an AI-based tracking system, idtracker.ai (*Romero-Ferrero et al., 2019*), to track and analyze individual fly's speed and direction at a temporal resolution of 1/60 s (*Figure 4A*, see *Video 1*). We were able to track 15–20 individual flies accurately and simultaneously through the entire course of an NGA trial (10 s). As shown in a representative tracked video from D45 (*Video 1*), female flies exposed to vmHT clearly showed an age-at-injury-dependent increase in the population of slow-moving flies, especially in the D31$_{Inj}$ cohort. In viewing NGA videos, it was also clear that though each trial is 10 s long, females took around 7 s to reach the top of the vial, whereas nearly all male flies reached the top of the vial by 4 s. Therefore, we elected to perform detailed quantitative analyses on climbing speeds and angles within the first 3 s after trial initiation, which better depicts the climbing responses to a startle stimulus.

Negative geotaxis represents a sensorimotor response that requires flies to first detect their position in respect to gravity and then move against the pull of gravity. Thus, injury-induced deficits could manifest in both the speed and angle of climbing. Speed analyses revealed that in both male and female flies vmHT exposure resulted in a shift in the distribution of speed to slower speeds. Female flies exhibited a progressively larger shift from D3$_{Inj}$, D17$_{Inj}$, to D31$_{Inj}$, and males only see this decrease in D17$_{Inj}$ and D31$_{Inj}$ cohorts, with D31$_{Inj}$ flies exhibiting the slowest speed (*Figure 4B*). Next, we quantified the directionality of each fly at every frame of each trial and presented the data in angular histograms (*Figure 4C*). Here, vertical direction to the top is defined by an angle of 0°, whereas angles of

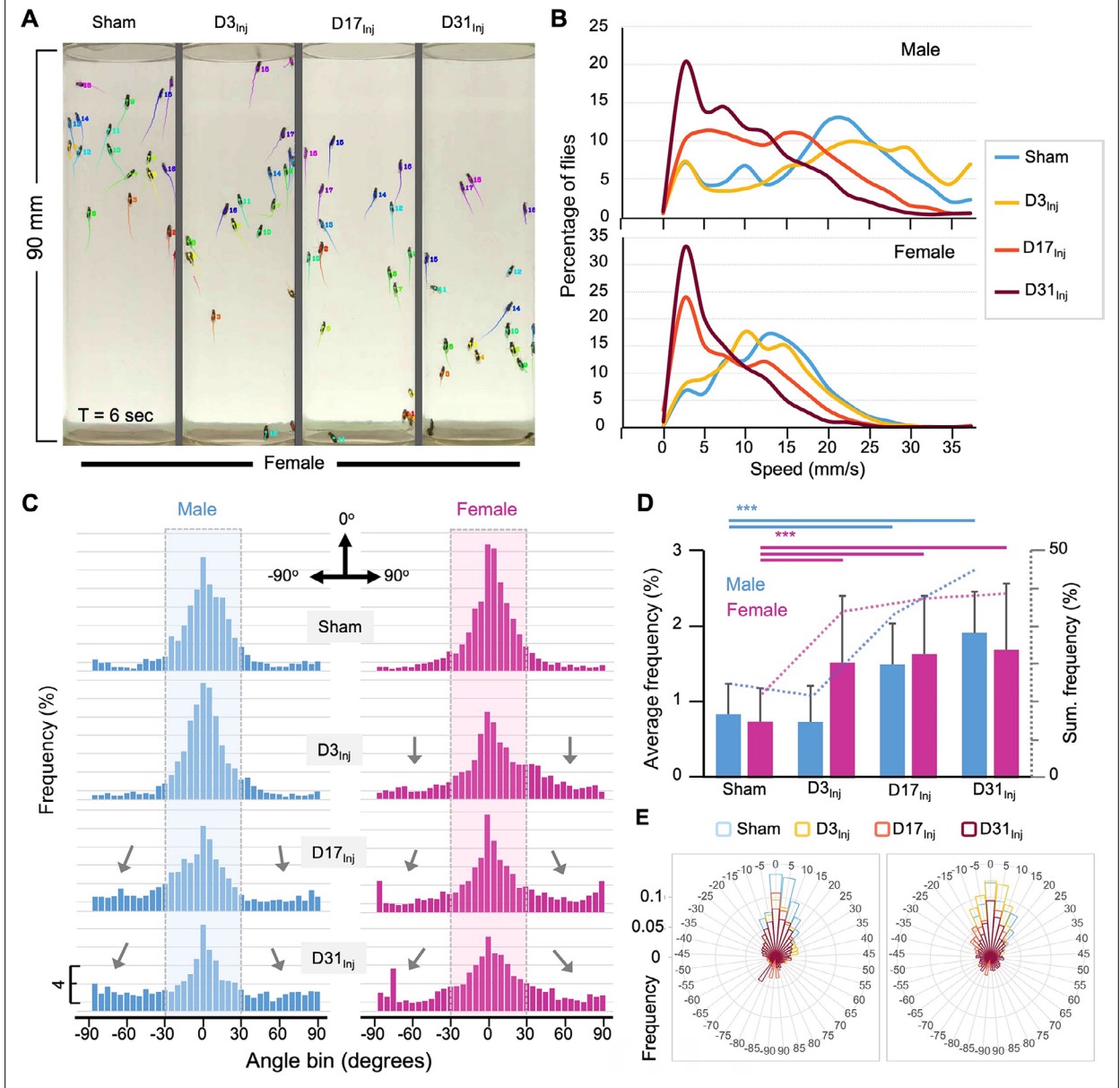

**Figure 4.** AI-assisted tracking and quantification of individual fly's behavior provides insight into defects in the speed and direction of movement. (**A**) Representative snapshot of idtracker.ai-generated video at T = 6 s. Videos used for analyses were from D45 flies (see *Video 1*). Each tail represents a fly's movement trajectory from the last 30 frames. Color is auto-assigned to individual flies in each trial. (**B**) Histogram plots of the climbing speed of individual flies at a temporal resolution of 1/60 s. The y-axis depicts the percentage of flies with a specific speed (x-axis). (**C**) Angular histograms showing the angle distribution of individual flies during the first 3 s of modified negative geotaxis assay (mNGA) trial. The highlighted sections represent normal fly directional orientation (between –30° and 30° with 0° as vertical). Arrows indicate the increased incidents of climbing angles outside the normal range. (**D**) Average percentages and accumulated percentages of flies with abnormal directional movement (<-30° or >30°) during the first three seconds of NGA trial. Two-sample *t*-tests, ***p<0.001. Error bar: se. (**E**) Rose plot depicting overall frequency distribution of fly orientation during the first 3 s of NGA trial.

–90° and 90° indicate that the fly is moving horizontally. We arbitrarily defined those flies moving with an angle between –30° and 30° as climbing normally in the vertical direction. vmHT exposure caused a significant portion of fly movement to fall outside the normal angular range, especially in flies injured at older ages (*Figure 4C and D*). Female flies in all injury groups exhibited significantly different and aberrant climbing angles compared to the sham, whereas for males, only D17$_{Inj}$ and D31$_{Inj}$ groups exhibited significant changes to climbing angles (*Figure 4E*). Together, the impairments in movement

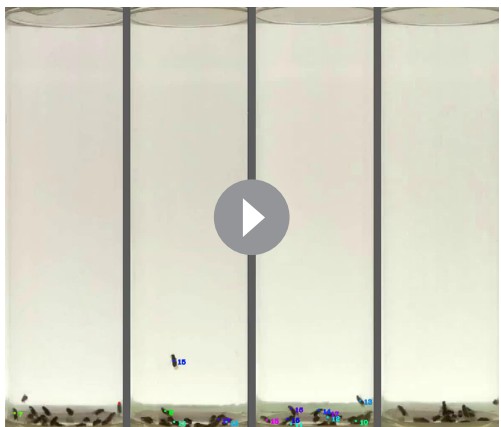

**Video 1.** AI-tracked climbing behaviors of female flies. Four groups of female flies on day 45 (D45) were subjected to the negative geotaxis assay and their climbing behaviors were recorded at a speed of 60 frames per second. From left to right, flies received no injury (sham) and flies received head trauma on days 3, 17, and 31 (D3Inj, D17Inj, D31Inj). For each group, the video recording was analyzed and tracked frame by frame by Idtracker.ai and the movement trajectories from the last 30 frames are superimposed on the video as color tails. Color is auto-assigned to individual flies in each trial. The dimensions of each column (each vial) are 28 mm (width) and 90 mm (height).

https://elifesciences.org/articles/97908/figures#video1

speed and movement direction suggest that both sensory and motor pathways were disrupted by vmHT exposure.

## Mating status determined female vulnerability to neurodegeneration following vmHT

Female flies are larger and about 30–40% heavier than males of the same age (*Figure 5A*). It is thus plausible that the elevated brain deficits observed in females had resulted from a larger impact force from vmHT. To test this possibility, we first reduced the impact force for female flies to match the impact force that the male flies experienced, which was accomplished by reducing the acceleration distance of the injury apparatus. We found that exposure to the reduced injury elicited similar behavioral deficits and brain pathology as exposure to regular vmHT (*Figure 5—figure supplement 1*). Since virgin females are similar in size and mass to mated females (*Figure 5A*), we next investigated the long-term responses of virgin females to vmHT. First, we found that like mated females, injury did not affect survival in virgin females regardless of age-at-injury (*Figure 5B*). Next, we quantified sensorimotor behavior and vacuolation at D45, which is when we consistently see neurodegenerative conditions in mated females of different injury groups. Unlike the mated females, virgin females displayed little injury-induced climbing deficits nor showed significant changes in brain vacuolation (*Figure 5C–E*). Together, these data suggest that the difference in body size/mass between male and female flies is unlikely the cause for the observed vulnerability to late-life neurodegeneration in mated females. It appears that mating somehow insidiously alters the female biology, rendering it more vulnerable to developing neurodegeneration after injury.

## Sex peptide is responsible for female vulnerability

Mating is known to induce a wide array of behavioral and physiological changes in female flies, collectively known as post-mating responses. Female post-mating responses are primarily triggered by the male accessory protein sex peptide (SP) (*Liu and Kubli, 2003*; *Chapman et al., 2003*) binding to SP receptors (SPRs) (*Yapici et al., 2008*) on a pair of sex peptide sensory neurons (SPSNs) (*Häsemeyer et al., 2009*; *Rezával et al., 2012*; *Yang et al., 2009*) within the female reproductive tract. We next investigated the role of SP signaling in injury-induced neurodegenerative conditions in mated females. First, female flies were mated to SP[0] males and then exposed to vmHT on D3, D13, or D31. mNGA and brain imaging on D45 revealed that these females behaved very similarly to the virgin females, with no lifespan reduction, sensorimotor deficits, or brain pathology due to vmHT at any age (*Figure 6A*). Next, pan-neuronal RNAi knockdown of SPRs in female flies yielded a similar result (*Figure 6B*). Like virgin females, these females exhibited no change in lifespan, negative geotaxis, or brain degeneration (*Figure 6B*). Finally, SPSN-specific RNAi knockdown of SPRs also abolished injury-associated deficits, regardless of age-at-injury (*Figure 6C*). It should be noted that the SPR RNAi line has been previously validated (*Yapici et al., 2008*) and its effectiveness in SPR knockdown is evident in female flies as they exhibit dramatically reduced egg laying and, importantly, lack typical post-mating behaviors (such as the rejection of male flies after the initial mating) observed in the wildtype-mated female flies. In fact, female flies with RNAi-mediated SPR knockdown behave identically to females mated with SP-null male flies, confirming the effective disruption of the SP-SPR signaling pathway.

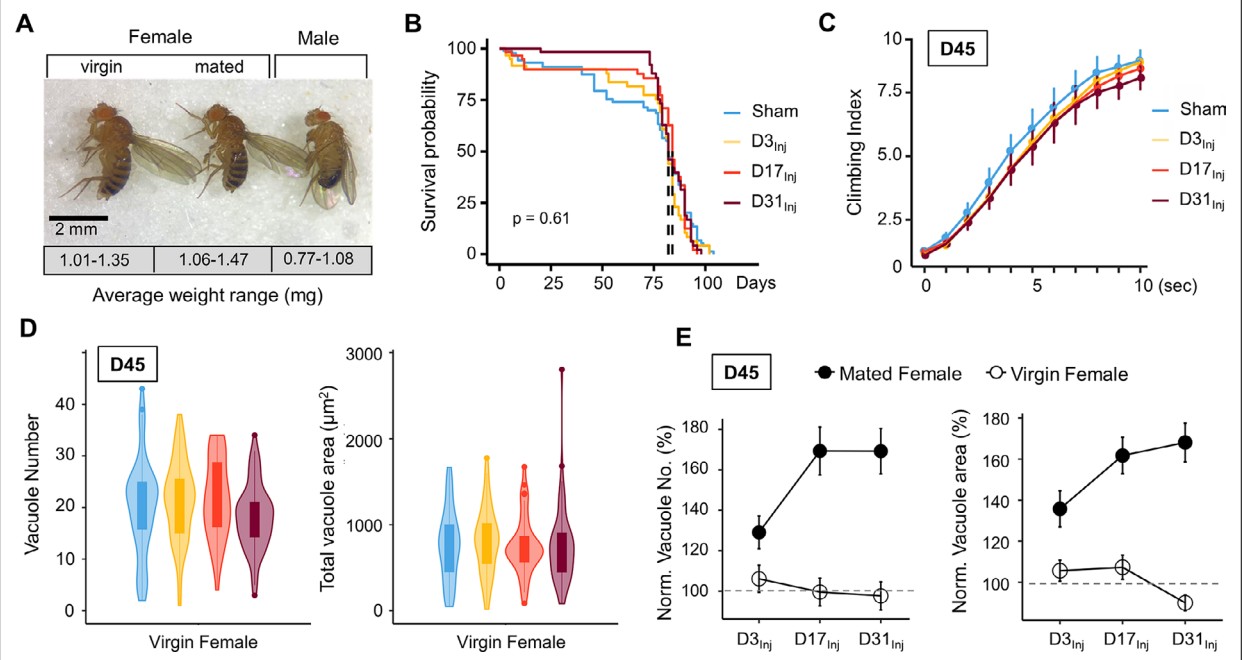

**Figure 5.** Virgin females do not exhibit neurodegenerative conditions regardless of age-at-injury. (**A**) Comparison of size and mass of virgin female, mated female, and male flies. Three separate trials of n > 100 in each condition. Total mass in each trial was averaged over the number of flies to generate a weight range for each condition. (**B**) Very mild head trauma (vmHT) resulted in no significant change to the lifespan of virgin females. Kaplan–Meier p-values were determined using the Mantel–Cox log rank test with Bonferroni correction. Total N = 271, n > 30 flies in each condition. (**C**) vmHT did not affect climbing behaviors of virgin females on D45 regardless of the age-at-injury. A total of nine videos were used for each condition: three experimental repeats of three trials, number of flies in each trial n ≥ 10. Repeated-measures ANOVAs were conducted to examine the effects of injury on climbing indices at each second (p>0.05). Error bar: ±se. (**D**) vmHT did not increase vacuole number or total vacuole area on D45 regardless of the age-at-injury. Total N = 224, n > 25 for each condition. Statistics: nonparametric Wilcoxon rank-sum tests. (**E**) Plots depicting differences in vacuole formation between virgin and mated females. Vacuole number and total vacuole area of each injury condition were normalized to the respective sham controls. Mated females exhibited much higher percentage of increase in vacuolation than virgin females. Error bar: ±se.

The online version of this article includes the following figure supplement(s) for figure 5:

**Figure supplement 1.** Exposure to reduced very mild head trauma (vmHT) on D3 elicited similar late-life behavioral deficits and brain pathology as exposure to regular vmHT.

Together, these data suggest that SP signaling through the reproductive pathway is responsible for the observed vulnerability to injury-induced neurodegeneration in mated females.

To further examine the relationship between SP signaling and injury-induced neurodegeneration, we performed a set of experiments in which virgin females were subjected to vmHT on D3, followed by mating with either wildtype or SP⁰ males for 24 hr on D10 (*Figure 7A*). We found that this post-injury mating scheme did not affect the lifespan of females that either mated with wildtype or SP⁰ males. Strikingly, post-injury mating with wildtype males effectively reintroduced injury-induced phenotypes, including increased vacuole formation and decreased climbing. However, post-injury mating with SP⁰ males did not significantly alter the virgin injury response (*Figure 7C and D*). Together, these results suggest that in females, pathways activated by SP signaling and those elicited by vmHT likely interact and synergize to accelerate the progression of neurodegeneration.

## RNA-seq analysis reveals sexually dimorphic responses to vmHT

To gain molecular insights into the sexually dimorphic responses to vmHT and the progressive development of late-life neurodegeneration, we conducted RNA sequencing on freshly dissected brain tissue from flies with and without vmHT exposure. Here, we focused on flies injured on day 3 after eclosion (D3$_{Inj}$) since under this injury condition only females mated with wildtype males exhibited behavioral and pathological deficits late in life (on D45, see *Figure 3*). Four distinct D3$_{Inj}$ groups were chosen for RNA extraction and sequencing: wildtype males, females mated with wildtype males, virgin females, and females mated with SP⁰ males (*Figure 8A*). We assessed transcriptomic changes in the

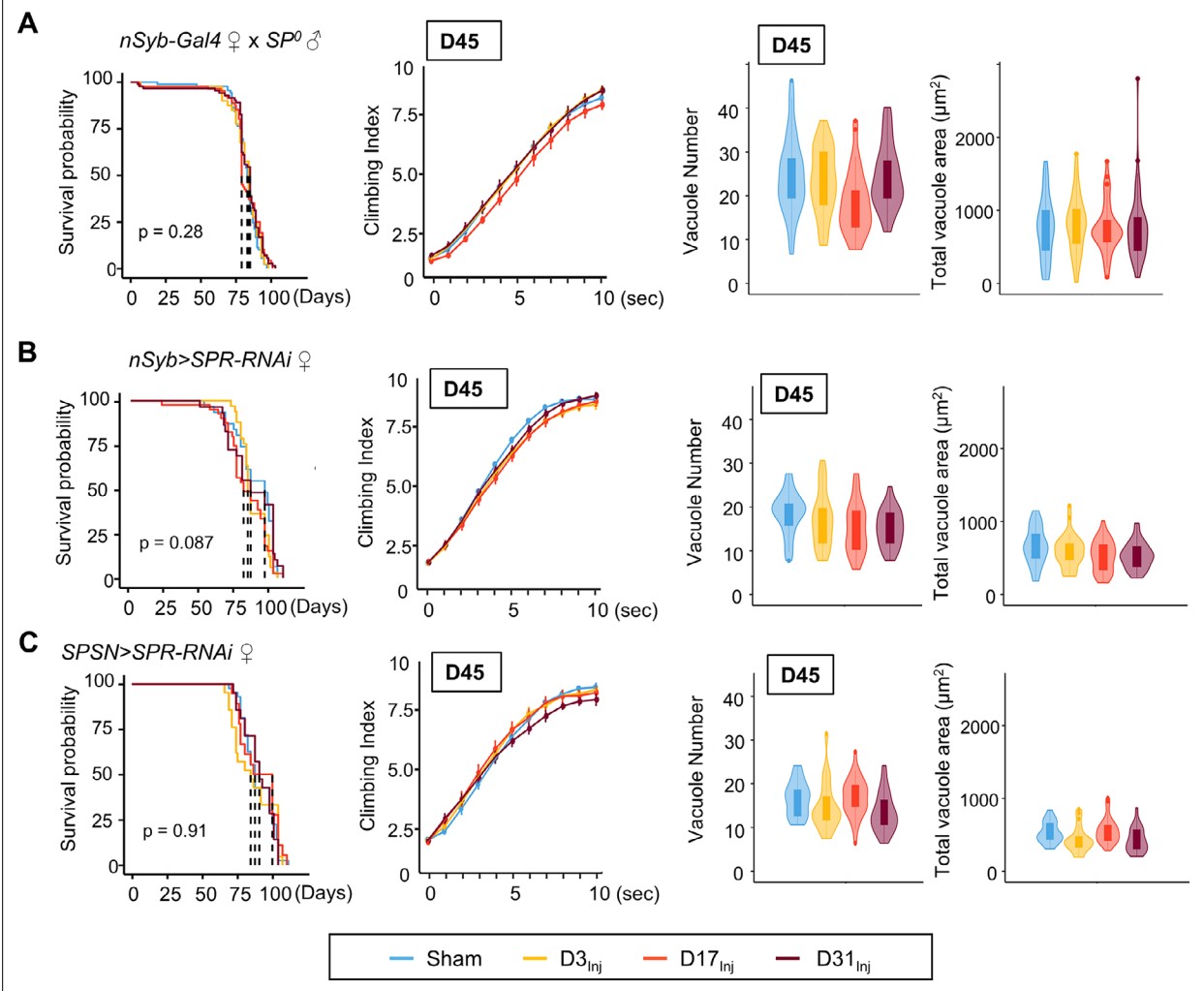

**Figure 6.** Eliminating SP signaling mitigates the emergence of neurodegenerative conditions in the female. (**A–C**) Survival curves, modified negative geotaxis assay (mNGA) quantification, and vacuole quantification of female flies mated to SP$^0$-males, females with pan-neuronal RNAi knockdown of SPR mated to wildtype males, and females with SPSN-specific knockdown of SPR mated to wildtype males (n > 30 in each condition). No difference was detected between sham and injured groups in all three genotypes. Error bar: ±se. Boxplots whiskers correspond to the maximum 1.5 interquartile range. Stats: nonparametric Wilcoxon rank-sum tests, p>0.05.

brain at two timepoints; 1 day post-injury (1 dpi) for acute changes and 6 weeks post-injury (6 wkpi) for chronic changes. Principal component analyses (PCA) revealed that sex accounted for a large portion of variance in sequencing dataset, but the effects of vmHT were not immediately clear in either time-points (**Figure 8B**). This observation reiterated the very mild nature of our injury paradigm. To further identify transcriptomic drivers of neurodegeneration after injury, we grouped the dataset by time post-injury, sex, and reproductive condition.

Compared to the respective sham groups, vmHT exposure resulted in significantly different sets of acute transcript changes between males and various female groups. D3$_{Inj}$ induced 18 upregulated and 58 downregulated transcripts in male brains at 1 dpi and 121 significant alterations (22 upregulated and 99 downregulated) in brains of wildtype-mated females (**Figure 8C**). However, we found a very few shared changes (14 genes) that are largely related to small molecule metabolic process (GO: 0044281) between male and female brains, suggesting sexually dimorphic responses to vmHT (**Figure 8D**, see **Supplementary file 1a** for the complete list of acute differentially expressed genes). Interestingly, we observed a much smaller number of transcriptomic changes in virgin females and SP$^0$-mated females (16 and 13, respectively) which minimally overlapped with injury-induced changes

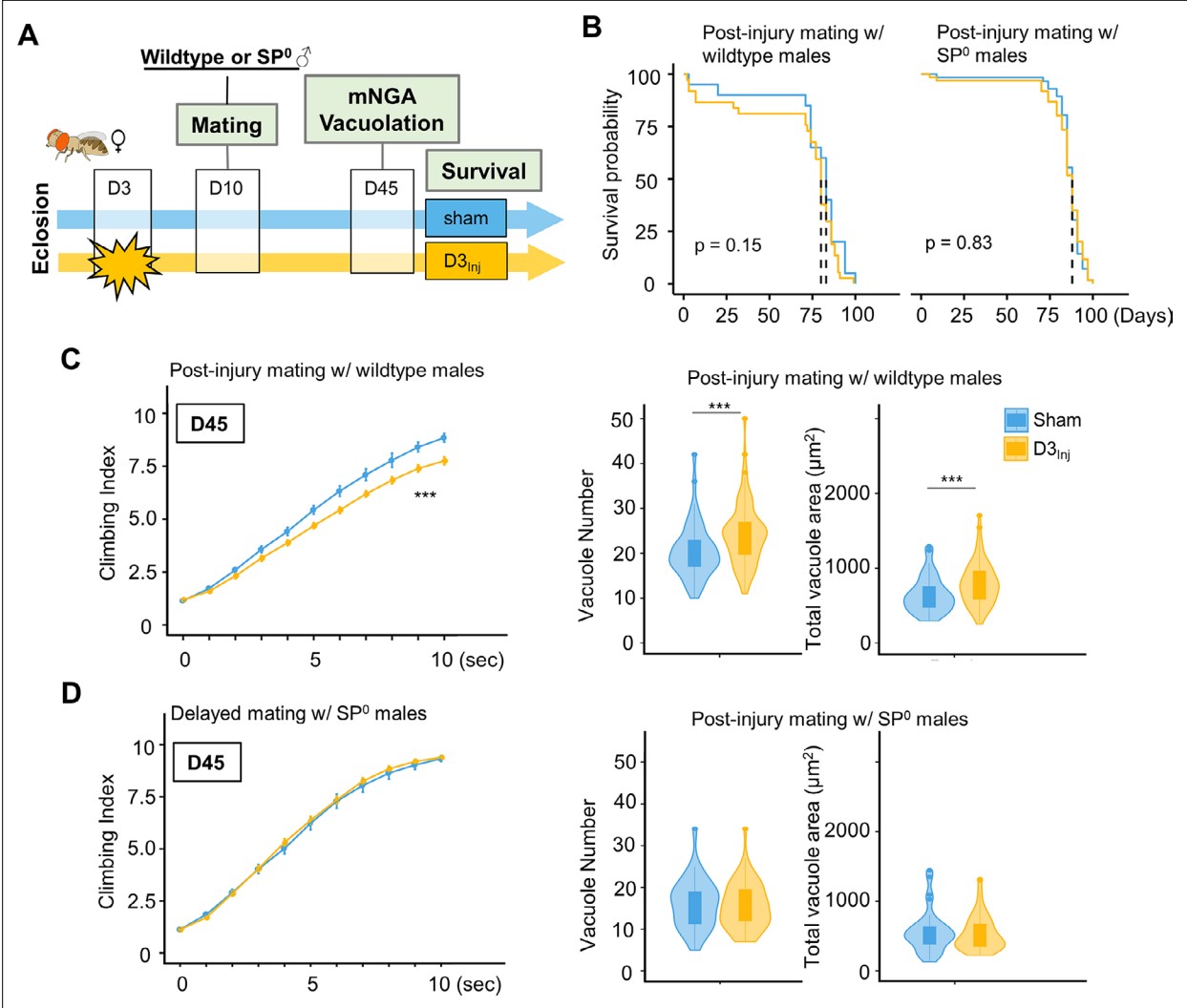

**Figure 7.** Introducing SP signaling to pre-injured virgin females reinstates neurodegenerative phenotypes. (**A**) Diagram of mating schematic and relevant behavioral and pathological assays. Virgin female flies were subjected to very mild head trauma (vmHT) on D3, whereas sham flies never received an injury. On D10, these females were exposed to either wildtype or SP0 males for 24 hr to allow mating. Modified negative geotaxis assay (mNGA) and brain vacuolation were assessed on D45. (**B**) No change in the lifespan was observed for females that were subjected to vmHT as virgin followed by post-injury mating with either male group. N = 182, n > 40 for each condition. (**C**) When assessed on D45, females subjected to post-injury mating with wildtype males exhibited significant sensorimotor deficits (***p=3.4e-15) and vacuole formation (vacuole number: p=0.0004536, vacuole area: p=0.0009563). A total of nine NGA videos were used for each condition: three experimental repeats of three trials, number of flies in each trial n ≥ 10. Error bar: ±se. Repeated-measures ANOVA and Bonferroni post hoc tests were conducted to examine the effects of injury on climbing indices at each second. For vacuole analyses, N = 149, n > 30 in each condition. (**D**) When assessed on D45, females subjected to post-injury mating with SP0 male flies did not elicit sensorimotor deficits and vacuole formation. A total of nine NGA videos were used for each condition: three experimental repeats of three trials, number of flies in each trial n ≥ 10. For vacuole analyses, N = 105, n > 20 in each condition.

in male and wildtype-mated female brains. Overall, data from the acute timepoint suggest that the immediate response to vmHT is sexually dimorphic and dependent on reproductive status.

Next, we analyzed differential transcriptomic changes of sham and D3$_{Inj}$ brains at the chronic timepoint (6 wkpi). Strikingly, only a small number of transcripts were found to undergo significant changes in each condition (*Figure 8E*). In males, D3$_{Inj}$ induced 1 upregulated and 10 downregulated gene transcripts on D45 compared to the sham group. In wildtype-mated females, D3$_{Inj}$ elicited eight significant alterations (two upregulated and six downregulated). Virgin female brains exhibited significant changes in only two transcripts (one upregulated and one downregulated), whereas SP0-mated female brains showed changes in 20 transcripts (15 upregulated and 5 downregulated) (for a list of all chronic differentially expressed genes, see *Supplementary file 1b*). We noted the lack of common

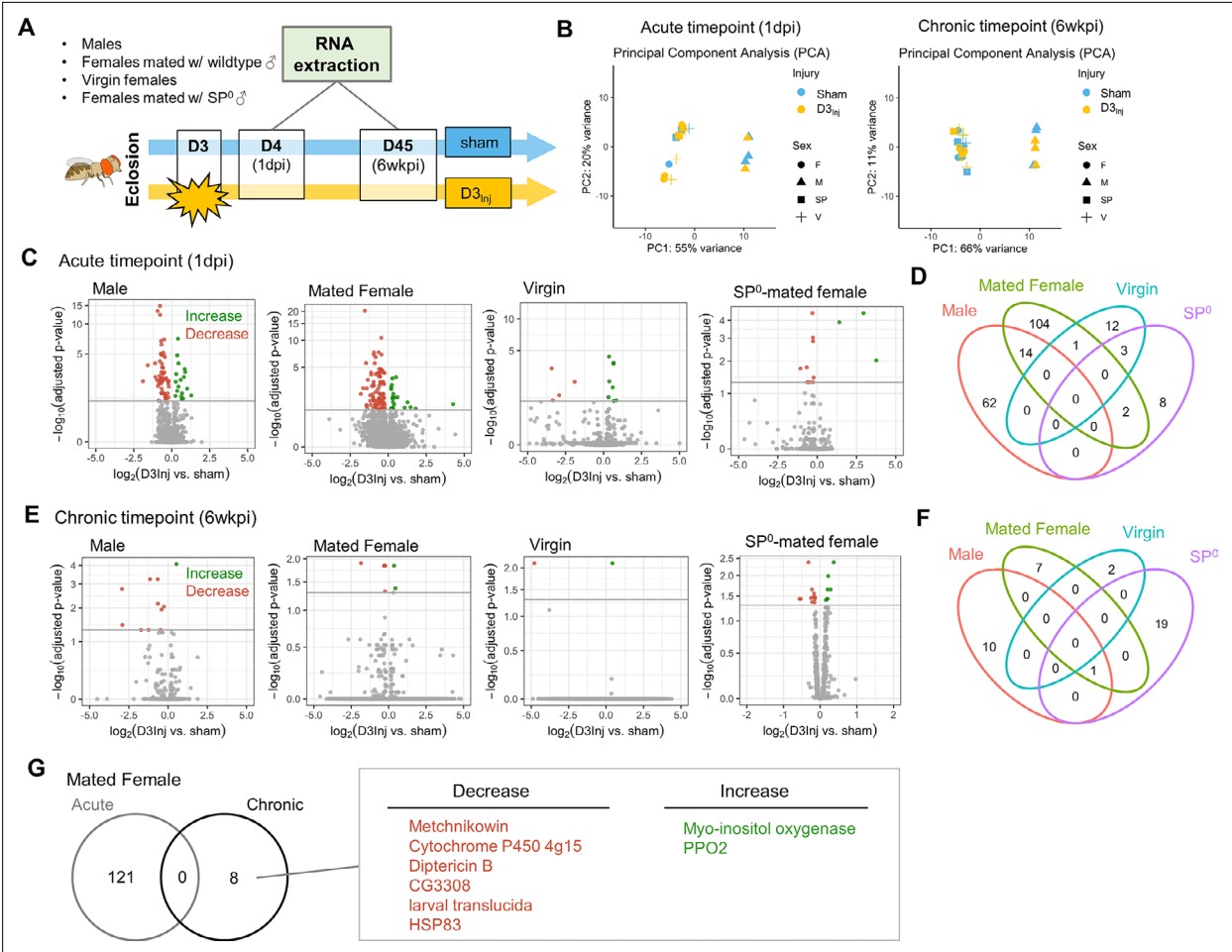

**Figure 8.** RNA sequencing reveals sexually dimorphic and reproductive status-dependent transcriptomic responses in D3$_{Inj}$ fly brains. (**A**) Timeline and design of RNA-seq experiment. Sham and D3$_{Inj}$ brains were collected at 1 day post-injury (1 dpi) and 6 weeks post-injury (6 wkpi), in four different conditions (wildtype males, females mated with wildtype males, virgin females, and females mated with SP$^0$ males), and in three biological replicates (n = 40 brains per condition per replicate). Total sample number n = 48. (**B**) Principal component analyses of samples at chronic and acute timepoints. (**C, E**) Volcano plots depicting gene expression at the acute and chronic timepoints. Upregulated genes are highlighted in green, downregulated genes are highlighted in red. (**D, F**) Venn diagram displaying the limited overlap of genes between the four conditions at the acute and chronic timepoints. (**G**) Venn diagram depicting no overlap between the acute and chronic gene changes in females mated to wildtype males. Upregulated genes are highlighted in green, downregulated genes are highlighted in red.

injury-induced changes between all four conditions, except for one transcript shared between males, wildtype-mated females, and SP$^0$-mated females (*Figure 8F*). This suggests that chronic transcriptomic profiles in response to vmHT were also sexually dimorphic and reproductive status dependent. Importantly, vmHT altered the expression of seven unique genes in wildtype-mated female brains. Three of these transcripts lie within immune system processes (GO:0002376), but the functions of the other four transcripts were not as clear. Specifically, mated females exhibited a decrease in the expression of two AMP genes, *Metchnikowin* and *Diptericin B*, both involved in the humoral immune response (*Hanson and Lemaitre, 2023*). We also observed an increase in the expression of *Prophenoloxidase 2*, whose activation is involved in the defense against pathogens (*Binggeli et al., 2014*). Furthermore, none of these genes were identified in the acute timepoint, suggesting that the altered expression of these genes may be involved in the late emergence of neurodegenerative conditions in mated females.

## Discussion

An emerging body of evidence suggests that early exposure to repetitive mild head trauma increases the risk of developing neurodegenerative conditions later in life. However, the underlying mechanisms remain unclear. In this study, we take advantage of the short lifespan and powerful genetic tools of *D. melanogaster*, in combination with our headfirst impact model (*Behnke et al., 2021b*), to investigate how early exposure to a very mild form of repetitive head trauma leads to late-life neurodegeneration. Previously, we showed that mild repetitive injuries can elicit degenerative phenotypes in both male and female flies, but female flies appear more vulnerable to the injuries (*Behnke et al., 2021b*). Our current study has further substantiated the previous findings by examining different age-at-injury cohorts and demonstrating their age-dependent and sexually dimorphic responses to a very mild form of repetitive head impacts. Furthermore, we discovered a strong link between SP signaling and injury-induced neurodegeneration in females. Compared to males, virgin females, and SP⁰-mated females, female flies mated to wildtype males were much more vulnerable to developing sensorimotor deficits and brain degeneration following injury exposure. This demonstrates, for the first time, the convergence and interaction of two supposedly separate pathways, namely the pathophysiological mechanisms elicited by mild head injury and the post-mating responses mediated by SP exposure. While activation of either one of these pathways was insufficient to generate neurodegenerative effects in females, their combination facilitated the emergence of quantifiable brain deficits. These findings provide an important foundation to further investigate the molecular and cellular mechanisms underlying sex differences in injury responses and neurodegeneration.

Sex-related differences have been documented in many medical conditions, including neurodegenerative disorders (*Vegeto et al., 2020*; *Cerri et al., 2019*; *Nebel et al., 2018*; *Podcasy and Epperson, 2016*) and neurotrauma of various severities (*Yue et al., 2019*; *Gupte et al., 2019*; *Rauen et al., 2021*). Sex differences include anatomical differences, sex gonadal hormones, and immunological responses, all of which presumably affect vulnerability to injury. In fact, there seems to be a growing consensus concerning sex-dependent effects in response to TBI, though clinical and preclinical findings are inconsistent. While female sex is more often associated with worse outcomes in human TBI studies, in preclinical TBI studies it is more frequently associated with better outcomes and neuroprotection after TBI (*Gupte et al., 2019*). Part of this disparity may be attributed to differences in injury severity and animal model, but the underlying drivers of sex differences in brain injury responses and neurodegenerative conditions remain to be fully elucidated. While sex differences in insects inherently differ from that in humans, there are some similarities associated with reproduction and hormone regulation. Therefore, it is likely that some of the fundamental mechanisms revealed in *Drosophila* may be conserved in humans.

Mating elicits two major types of changes in female flies; increase in egg-laying and reduction in receptivity to mating (*Liu and Kubli, 2003*; *Chapman et al., 2003*). Sperm alone can induce a short-term post-mating response, but a sustained response (>7 days) requires SP (*Liu and Kubli, 2003*). SP signaling via SPR activation is believed to switch the female to a state that maximizes reproduction (*Yapici et al., 2008*), which is considered evolutionarily beneficial. However, SP exposure can jeopardize female survival by decreasing daytime sleep (*Isaac et al., 2010*; *Garbe et al., 2016*), altering the immune system (*Schwenke and Lazzaro, 2017*; *Short and Lazzaro, 2010*; *Short et al., 2012*; *Peng et al., 2005*), and increasing female susceptibility to age-dependent tumors (*Regan et al., 2016*). To our knowledge, the current study is the first to directly show that SP-signaling can elevate vulnerability to injury-induced neurodegeneration in female flies. While mammals do not have SP, reproduction in females is known to dramatically alter metabolism, immune system, hormones, and neurobiology, all of which are vital for maintaining pregnancy. A connection between increased risk for neurodegeneration and hormonal changes associated with reproduction has been suggested in humans and other mammals (*Xiong et al., 2022*; *Gilsanz et al., 2019*; *Jang et al., 2018*). There are also contradictory findings, where female sex hormones are found to be neuroprotective (*Ha et al., 2007*; *Song et al., 2020*). These findings highlight the need for further animal studies with better controls of hormonal and reproductive cycles.

The current study also examines how aging affects response to mild head injury. In humans, age is one of the strongest outcome predictors for complications following head trauma, including mild trauma (*Moretti et al., 2012*; *Jacobs et al., 2010*; *Bittencourt-Villalpando et al., 2021*). Though the cause of injury may differ, older age is consistently associated with an increased incidence of head

trauma, a slower overall recovery process, as well as greater morbidity and mortality following injury (*Abdulle et al., 2018*; *Mosenthal et al., 2004*). Consistent with these observations, we show that flies exposed to vmHT at older ages developed neurodegenerative deficits at a much faster rate than flies injured at younger ages. Our finding that aging accelerates injury-induced neurodegeneration is also supported by findings in humans and other animal models (*Doust et al., 2021*; *Rowe et al., 2016*; *Ojo et al., 2013*), suggesting conserved pathways underlying aging-induced vulnerability. Previous studies have suggested that mitochondrial function (*Cui et al., 2012*; *Chistiakov et al., 2014*) and immune function (*Weyand and Goronzy, 2016*; *Haynes, 2020*) decline with aging. Given that mitochondrial dysfunction (*Cheng et al., 2012*; *Kim et al., 2017*) and compromised immune responses *Needham et al., 2019*; *Verboon et al., 2021* have also been implicated in TBI-induced brain dysfunction and degeneration, future work using our mild injury paradigm should investigate their contributions to aging-increased vulnerability and the development of neurodegeneration in response to head trauma.

Sex differences in aging are conserved across species. In line with this, we demonstrated sex differences in aging-associated vulnerability. While increasing age-at-injury in mated females is generally associated with more severe neurological deficits, females injured on D17 exhibit similar degrees of climbing deficits and vacuolation to those injured on D31, suggesting a possible plateau (*Figure 3A*). On the other hand, male flies are very resilient to vmHT at younger ages (D3 and D17), displaying little to no phenotypes even on D45 (*Figure 3A*). However, at the advanced age of 31 days post-eclosion, exposure to vmHT drastically accelerates neurodegeneration in the male brain, resulting in deficits similar to that of mated females of the same injury group (*Figure 3B and E*). As flies age, male flies seem to show a larger decline in resistance to injury-induced neurodegeneration than females. This finding is supported by work studying sex differences in immune senescence (*Kubiak and Tinsley, 2017*). In males, this is believed to occur principally due to age-related deterioration in barrier defenses, whereas in females this largely manifests in decreases in innate immunity. Older males may have a weaker exoskeletal defense system, making them more vulnerable to the effects of injury.

To gain some insights into the molecular mechanisms underlying the sexually dimorphic, late-life emergence of neurodegenerative conditions after early exposure to vmHT, we surveyed the transcriptomic landscapes of fly brains with and without vmHT exposure. We were specifically interested in identifying genes whose changes in expression might underlie the observed sex differences between male and mated females, as well as between females of varying reproductive status. Across all conditions and timepoints, we found that the injuries (regardless of age) elicited a very small number of differentially expressed genes that skewed toward decrease in expression. This was initially surprising, because previous *Drosophila* TBI work using the same RNA extraction protocol and sequencing company had reported large numbers of both immediate and lasting gene activation (*Byrns et al., 2021*). When we compared sham males and females at the acute timepoint, we found robust sex-specific gene expression (data not shown) and distinct clustering by sex in our PCA (*Figure 6B*). This confirmed the validity of our data and subsequent analyses and suggested that the small number of differentially expressed genes was likely due to the mild nature of our injury paradigm. Furthermore, it is possible that not all flies experienced increased neurodegeneration following the mild injuries, which could mask transcriptomic changes due to injury-induced neurodegeneration. Unlike other work utilizing a more severe injury paradigm (*Byrns et al., 2021*), we did not find any acute gene changes that persisted until the very chronic timepoint of 6 weeks following injury, suggesting that mild injury may induce dynamic transcriptomic changes throughout a fly's lifespan. Additionally, cellular mechanisms that directly contribute to the development of neurodegenerative conditions may not be immediately active following injury; instead, they may be revealed by aging. To better understand the connections and interactions between acute and chronic changes in the transcriptome, future studies may further characterize these dynamic changes at multiple timepoints throughout the fly's lifespan.

Previous work using a mild injury model has shown that AMP expression is elevated in the brain immediately following injury (in the first 12 hr), returned to baseline at 24 hr post-injury, and then elevated again 1 week post-injury (*Barekat et al., 2016*). This suggests that persistent elevation of AMP expression levels and the activation of the innate immune system at a chronic timepoint contributes to neurodegeneration. For both mated females and males, our results comparing injured and sham flies at the acute timepoint (1 dpi) do not show any elevation of AMP or AMP-like genes,

which is congruent with previous data. A small percentage of immune-related gene changes were identified at this time, but these changes were all biased towards an injury-induced suppression. However, at 6 weeks post-injury, we find that the injury paradigm overwhelmingly depressed AMP and AMP-like gene expression in both male and mated female brains but not in virgin and SP-null mated female brains. Interestingly, injury-induced suppression of *Metchnikowin* and *Diptericin* and increased expression of *PPO2* observed at the chronic timepoint is correlated with female-specific late-life behavioral deficits and brain pathology. This finding is supported by a recent report that mated female flies exhibited decreased markers of oxidative stress following trauma, which could be due to immune suppression following a TBI-induced immune challenge (*Jones et al., 2024*). Additionally, work comparing bacterial infection in virgin, SP⁰-mated, and wildtype-mated females suggests that mating and particularly egg-laying reduce the female's overall ability to defend against bacterial infections, leading to decreased expression of AMP genes after infections (*Short et al., 2012*). This process is mediated by increases in Juvenile Hormone (*Schwenke and Lazzaro, 2017*), which is known to suppress innate immune processes (*Flatt et al., 2008*). SP exposure also acutely activates the Toll-like receptor and immune deficiency (Imd) pathways in females, altering levels of *Metchnikowin* and *Diptericin* (*Peng et al., 2005*). Furthermore, even one instance of mating is sufficient to induce chronic immunosuppression (*Gordon et al., 2022*). These changes may compound injury-induced changes to disrupt immune homeostasis, leading to observed immunosuppression in mated females. At an advanced age, a decreased expression of AMPs suggests that the mated female fly's innate defense may be worsened by injury, making them unable to mount a sufficient immune response to pathological insults and therefore more susceptible to infections and disease.

Our finding of the decreased expression of AMP genes at this very chronic timepoint of 6 weeks post-injury, in combination with other groups' findings of short-term elevation of AMP genes, also suggests a temporally dynamic gene expression pattern in response to head injury. Though immune suppression in mated females seems to contribute to neurodegenerative phenotypes, it is not immediately clear why suppression of a different set of AMP genes was not sufficient for neurodegenerative phenotypes in males. Future work can utilize males injured at older ages or with a more severe paradigm to understand male-specific transcriptome changes that contribute to injury-induced neurodegenerative conditions. Finally, though our RNA-seq data is informative, we acknowledge that there is limited correlation between mRNA and protein levels. Clearly, future studies are needed to dissect the genetic components and molecular players involved in the sex-different development of neurodegenerative conditions after mild head trauma.

## Materials and methods

### Fly husbandry

Flies were maintained at 25°C, with 60% humidity on a 12 hr:12 hr light:dark cycle and kept in vials containing fresh fly media consisting of cornmeal, yeast, molasses, agar, and *p*-hydroxy-benzoic acid methyl ester. Vials were changed every 2–3 days to ensure fresh food and minimize flies getting stuck to wet food. The following stocks were used: *w¹¹¹⁸* (BDSC, 5905), *neuronal synaptobrevin-GAL4* (*nSyb-GAL4*) (generously provided by Dr. T. Ngo from the Rubin Lab at Janelia Research Campus), SP null mutant line (0325/TM3, *Sb, ry*), SP deficiency line (Δ130/TM3, *Sb, ry*) (*Liu and Kubli, 2003*) (both generously shared by Dr. M. Wolfner at Cornell University), *SPSN-splitGAL4* (*Wang et al., 2020*) (generously provided by Dr. K. Koh at Thomas Jefferson University), and *UAS-SPR RNAi* (BDSC, 66888). SP⁰ males (0325/Δ130) were generated by crossing the SP null mutant line to the deficiency line. Pan-neuronal and SPSN-specific knockdown of SPR was achieved by crossing *UAS-SPR RNAi* females to *nSyb-GAL4* and *SPSN-splitGAL4* males, respectively. Unless otherwise noted, flies used were progeny from *nSyb-GAL4* males crossed to *w¹¹¹⁸* females. Virgin females and males were collected within 8 hr of eclosion and kept in separate vials. For mated flies, 1 day after collection, female and male flies were given 24 hr to mate and separated again after. Flies were handled meticulously without $CO_2$ after separating by sex to negate anesthetic effects.

### Injury paradigm and apparatus

Flies were injured in accordance with the protocol detailed in *Behnke et al., 2021b*. Briefly, multiple flies were contained within a custom plastic injury vial designed to fit within an injury rig consisting of a

cradle connected to a pulley system with a counterweight. To ensure consistency across injuries, each injury trial consisted of pulling the cradle downward to the base of the injury rig, followed by lightly tamping the cradle three times for all the flies to fall to the bottom of the vial before releasing it for upward acceleration. When the cradle and injury vial stopped at the top, the momentum of the fruit flies continued to propel them upward to the glass ceiling, where they sustained headfirst impacts. In this study, we modified the injury paradigm such that it consisted of one session of 15 impacts spaced 10 s apart (referred to as vmHT). Flies were injured at 3 days ($D3_{Inj}$), 17 days ($D17_{Inj}$), or 31 days post eclosion ($D31_{Inj}$). Sham flies (control) were placed into the injury vials for the duration of an injury session (2.5 min) but did not receive any injuries. For subsequent data analyses, we combined all sham groups within each experiment as we observed no behavioral or pathological differences between them.

A subset of flies was subject to an eye-penetrating injury as a positive control for TUNEL staining. Here, flies were anesthetized on a $CO_2$ pad while a fly pin penetrated one side of the eye and into the brain. Flies were given 1 day to recover before fixation and subsequent immunofluorescent staining. Dead flies were excluded from subsequent analyses.

## Video-assisted startle-induced climbing assay

Startle-induced climbing behavior was assessed using an mNGA. Climbing was assessed at the following timepoints after injury: 90 min, 1 day, 1 week, and weekly until D45. Flies were placed in plastic vials with a soft bottom consisting of 5% agar. Custom 3-D-printed caps wrapped in parafilm were used to contain the flies. Up to four vials were assayed at the same time, using a custom 3-D polylactic acid (PLA) printed rig. A lightboard was placed behind the apparatus to enhance background contrast for the video recordings. For each trial, the rig was lightly tamped three times, and fly movement was recorded using a Panasonic HC-V800 digital video recorder at 60 frames per second. Three trials were conducted for each video, spaced 1 min apart. Using FIJI, videos were cropped to fit individual vials, and then trimmed to the first 601 frames (10 s) after flies fell to the bottom. The 10 s videos then underwent manual fly behavior tracking or idTracker.ai and subsequent analyses. All testing took place between ZT 3 and ZT 8 (ZT, Zeitgeber time; lights are turned on at ZT 0 and turned off at ZT 12) and testing occurred between 20 and 23°C under laboratory lighting conditions.

Manual fly tracking was done using FIJI. Briefly, using the segmentation tool, the length of each vial was segmented into 10 equal bins and the number of flies in each bin was recorded at every second (every 60 frames), starting at the first frame. A CI for each second was calculated using the sum of the number of flies in each bin multiplied by the bin number and then divided by the total number of flies in the trial. Repeated-measures ANOVAs were used to compare the effects of the various ages-at-injury conditions on CIs for each second of the trials and post hoc tests were performed using the Bonferroni method.

Automated AI-powered tracking of fly negative geotaxis behavior was performed using idtracker. ai (*Romero-Ferrero et al., 2019*), a software utilizing a deep-learning algorithm that permits simultaneous positional tracking of multiple flies at high temporal resolution. 10 s NGA videos were processed in an Anaconda environment using a Dell workstation equipped with an NVIDIA Quadro RTX5000 graphic card. Tracking results were manually validated using the validate tool provided by idtracker. ai before generating fly coordinates in regard to the XY plane. Superimposed video clips with the speed tail for each fly were generated using the video tool from the idtracker.ai package. Python scripts based on Trajectorytools from idtracker.ai were used to determine the speed and directional orientation frame-by-frame in addition to climbing trajectories for each fly. Data was imported into Microsoft Excel and histogram plots were generated using Excel. Statistical analyses were performed using two-sample *t*-tests.

## Immunohistochemistry

Immunohistochemistry and image analyses were performed as described in our previous publication (*Behnke et al., 2021a*). Whole flies were fixed for 3 hr in 4% paraformaldehyde (PFA) in phosphate-buffered saline (PBS) containing 0.5% Triton-X (PBS-T) at room temperature with nutation. Flies were next rinsed four times for 15 min each with PBS-T and brains were subsequently dissected in PBS. Brains were permeabilized overnight in 0.5% PBS-T at 4°C with nutation then blocked in 5% normal goat serum (NGS) in PBS-Triton for 90 min at room temperature with nutation. Brains were stained

with Hoechst 33342 solution (1:1000; Thermo Scientific 62249) and Alexa Fluor 594 (AF594) phalloidin (1:400, Invitrogen A12381). Regions devoid of Hoechst and phalloidin signal that are not associated with physiological structures such as the esophagus were considered vacuoles. For TUNEL staining, brains were incubated with the mixture of two solutions in the In Situ Cell Death Detection Kit, Fluorescein (Roche-11684795910) at 37°C for 1 hr. For structural analyses of PDF neurons and mushroom bodies, brains were incubated with 5% NGS and primary antibodies (1:100 DSHB mouse anti-PDF C7 or 1:50 mouse anti-Fasciclin II 1D4) in 0.5% PBS-T for 2 days at 4°C with nutation. Following primary antibody staining, brains were then washed four times in 0.05% PBS-T and then incubated with 5% NGS and secondary antibody (1:400 Alexa 594 goat anti-mouse IgG, Invitrogen A11032) in 0.5% PBS-T for 2 days at 4°C with nutation. Stained brains were rinsed 4 × 15 min with PBS-T, followed by one wash in PBS for 90 min and then mounted on glass slides within SecureSeal Imaging Spacers (Grace Bio-Labs, Bend, OR) containing SlowFade Gold Antifade mounting medium (Life Technologies, Carlsbad, CA S36937). Whole-brain imaging at 1 μm steps was performed using an Olympus FV1000MPE two-photon system equipped with a water immersion objective XLPLN25XWMP (working distance of 2 mm and N.A. of 1.05). For TUNEL staining, imaging was performed using a confocal microscope (Nikon C2+confocal system based on a Ti2-E microscope) at 1 μm steps. Image analysis was performed using ImageJ (Fiji) software. The number and size of vacuoles were recorded and statistically assessed using nonparametric Wilcoxon rank-sum tests and post hoc tests using the Bonferroni method.

## RNA extraction and sequencing

Injured and sham flies were injured at 3 days post-eclosion. At 1 day and 6 weeks post-injury, brains from males, wildtype-mated females, virgin females, and $SP^0$-mated females were collected, resulting in 16 conditions, each condition containing three independent cohorts of n = 40 flies. RNA from brain tissue was extracted by TRIzol/Chloroform extraction (Thermo Fisher, 15596026) and further clean-up with the Zymo DNA Clean & Concentrator-5 kit (D4004) with on-column DNase treatment (Thermo Fisher, AM1907). Once obtained, RNA samples were transferred on dry ice to Admera Health for library preparation and sequencing. Total RNA integrity was checked by BioAnalyzer. QIAseq FastSelect rRNA Fly+NEB Ultra II Directional kit was used to deplete rRNA and prepare stranded cDNA libraries. Samples were sequenced on NextSeq High Output Flow Cell (Illumina) for 150 cycles to generate a total of 60 M paired-ended, 101 base-pair reads.

## Bioinformatic analyses

Following sequencing, raw FASTQ reads passing QC filter (Q > 30) were obtained from BaseSpace. Analyses were conducted on the Galaxy web platform using the public server at https://usegalaxy. org/ (*Afgan et al., 2018*) and then on R on a local machine. Quality control on the data was performed using FastQC (*Andrews, 2010*), and data was subsequently mapped using RNA STAR (*Dobin et al., 2013*) and the Ensembl (*Cunningham et al., 2022*) gene annotation for *Drosophila melanogaster* reference genome (dm6). Mapping results were examined using IGV (*Robinson et al., 2011*), IdxStats from the Samtools suite (*Danecek et al., 2021*), and Gene Body Coverage tool and Read Distribution from the RSeQC tool suite (*Wang et al., 2012*). Mapped reads were then assigned to exons/genes and tallied using featureCounts (*Liao et al., 2014*). To enable inter-sample read count comparisons, count normalization and differential expression analysis were conducted using DESeq2 (*Love et al., 2014*) with models that incorporate sex, age, and injury conditions. Functional enrichment analyses of any differentially expressed genes were conducted using enrichGO from clusterProfiler 4.0 (*Wu et al., 2021*). Manual searches and annotations of genes were performed using FlyBase (*Gramates et al., 2022*). RNAseq raw and processed data are available at Gene Expression Omnibus (GEO, access number: GSE274136).

## Data reporting and statistical analysis

No statistical methods were used to determine sample sizes but are consistent with sample sizes of those generally employed in the field. All experiments contained three biological replicates, unless otherwise noted. All flies in each vial were administered the same treatment regimen. For each experiment, the experimental and control flies were collected, treated, and tested at the same time. For behavioral and pathological data, statistical analyses for comparisons were as described above. All

statistical analyses were performed using R packages (R v4.2.2, [rstatix]). p-values are indicated as follows: ***p<0.001; **p<0.01; *p<0.05.

## Acknowledgements

We thank Dr. Joseph A Behnke for his help and input throughout this research project. We also thank Dr. Kenneth Myers and Katherine Hardin for their feedback on the manuscript. This research was supported in part by a pilot grant to JQZ from the NIH-funded Emory Specialized Center of Research Excellence in Sex Differences (U54AG062334) and a Diana Jacobs Kalman Scholarship from the American Federation for Aging Research (2022) to CY.

## Additional information

### Funding

| Funder | Grant reference number | Author |
| --- | --- | --- |
| National Institutes of Health | U54AG062334 | James Q Zheng |

The funders had no role in study design, data collection and interpretation, or the decision to submit the work for publication.

### Author contributions

Changtian Ye, Conceptualization, Data curation, Formal analysis, Validation, Investigation, Visualization, Methodology, Writing - original draft, Project administration, Writing – review and editing; Ryan Ho, Formal analysis; Kenneth H Moberg, Supervision, Writing – review and editing; James Q Zheng, Conceptualization, Data curation, Software, Formal analysis, Supervision, Funding acquisition, Validation, Investigation, Visualization, Methodology, Project administration, Writing – review and editing

### Author ORCIDs

Changtian Ye ⓘ https://orcid.org/0000-0002-5381-410X
Kenneth H Moberg ⓘ https://orcid.org/0000-0002-9820-5543
James Q Zheng ⓘ https://orcid.org/0000-0002-8093-422X

Reviewer #1 (Public Review): https://doi.org/10.7554/eLife.97908.3.sa1
Reviewer #2 (Public Review): https://doi.org/10.7554/eLife.97908.3.sa2
Reviewer #3 (Public Review): https://doi.org/10.7554/eLife.97908.3.sa3
Author response https://doi.org/10.7554/eLife.97908.3.sa4

## Additional files

### Supplementary files

• Supplementary file 1. Summary of the RNAseq data on transcriptomic changes and all the statistical comparison results. (a) List of all differentially expressed genes acutely after vmHT exposure. (b) List of all differentially expressed genes at the chronic time point (day 45) after vmHT exposure on day 3. (c) List of statistical comparisons and p-values.

• MDAR checklist

### Data availability

RNA sequencing data have been deposited in GEO under the access code GSE274136.

The following dataset was generated:

| Author(s) | Year | Dataset title | Dataset URL | Database and Identifier |
|---|---|---|---|---|
| Ye C, Ho R, Moberg KH, Zheng JQ | 2024 | Adverse Impact of Female Reproductive Signaling on Age-Dependent Neurodegeneration After Mild Head Trauma in Drosophila | https://www.ncbi.nlm.nih.gov/geo/query/acc.cgi?&acc=GSE274136 | NCBI Gene Expression Omnibus, GSE274136 |

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
