## [Editor Report · eLife assessment]

The authors have presented an interesting set of results showing that female sex peptide signaling adversely affects late-life neurodegeneration after early-life exposure to repetitive mild head injury in *Drosophila*. This **fundamental** work substantially advances our understanding of how sex-dependent response to TBI occurs by identifying the sex peptide and the immune system as modulators of sex differences. The evidence supporting the conclusions is **compelling** with rigorous inclusion of controls and appropriate statistics.

---

## [Referee Report · Reviewer #1 (Public Review)]

Summary:

In this manuscript the authors use the model organism *Drosophila* to explore sex and age impacts of a TBI method. They find age and sex differences: older age is susceptible to mild TBI and females are also more susceptible. In particular, they pursue a finding that virgin vs mated females show different responses: virgins are protected but mated females succumb to TBI with climbing deficits. In fact, virgin females compared to mated females are largely protected. They discover this is associated with exposure of the females to Sex Peptide in the reproductive neurons of the female reproductive tract. When they extend to RNAseq of brains, they show that there are very few genes in common between males, mated females, virgins and females mated with males lacking sex peptide. But what the few chronic genes associated with mated females seem associated with the immune system. These findings suggest that mated females have a compromised immune system, which might make them more vulnerable. In a bigger context, these findings point to the idea that the life status of the animal/individual may have an important impact on the outcome of a TBI - here illustrated by the differential state of virgin vs mated females in *Drosophila*.

Strengths:

This is an interesting paper that allows a detailed comparison of sex and age in TBI which is largely only possible in such a simple model, where large numbers and many variations can be addressed. Overall the findings are interesting.

Weaknesses:

Although the findings beyond Sex peptide are observational, the work sets the stage for more detailed studies to pursue the role of the genes they find by RNAseq and whether for example, boosting the innate immune system would protect the mated females, among other experiments.

---

## [Referee Report · Reviewer #2 (Public Review)]

Summary:

In this manuscript, the authors use the *Drosophila* model system to study the impact of mild head trauma on sex-dependent brain deficits. They identify Sex Peptide as a modulator of greater negative outcomes in female flies. Additionally, they observe that increased age at the time of injury results in worse outcomes, especially in females, and that this is due to chronic suppression of innate immune defense networks in mated females. The results demonstrate a novel signaling pathway that promotes age- and sex-dependent outcomes after head injury.

Strengths:

The authors have modified their previously reported TBI model in flies to mimic mild TBI, which is novel. Methods are explained in detail, allowing for reproducibility. Experiments are rigorous with appropriate statistics. A number if important controls are included. The work tells a complete mechanistic story and adds important data to increase our understanding of sex-dependent differences in recovery after TBI. The Discussion is comprehensive and puts the work in context of the field.

Weaknesses: None

The authors answered the following concerns, and I have no other concerns:

A very minor weakness is that exact n values should be included in the figure legends. There should also be confirmation of knockdown by RNAi in female flies either by immunohistochemistry or qRT-PCR if possible.

---

## [Referee Report · Reviewer #3 (Public Review)]

Summary:

In this manuscript, the authors used a *Drosophila* model to show that exposure to repetitive mild TBI causes neurodegenerative conditions that emerge late in life and disproportionately affect females. In addition to the well-known age-dependent impact, the authors identified Sex Peptide (SP) signaling as a key factor in female susceptibility to post-injury brain deficits.

Strengths:

The authors have presented a compelling set of results showing that female sex peptide signaling adversely affects late-life neurodegeneration after early-life exposure to repetitive mild head injury in *Drosophila*. They have compared the phenotypes of adult male and female flies sustaining TBI at different ages, and the phenotypes of virgin females and mated females, (2) compared the phenotypes of eliminating SP signaling in mating females and introducing SP-signaling into virgin females, (3) compared transcriptomic changes of different groups in response to TBI. The results are generally consistent and robust.

Weaknesses:

The authors have made their claims largely based on assaying climbing index and vacuole formation as the only indicators of late-life neurodegeneration after TBI. Furthermore, it is also really surprising to see so few DEGs even in wild-type males and mated females and to see that none of DEGs overlap among groups or are even related to the SP-signaling. The authors state that the reason is their TBI is very minor. It is critical to independently verify their RNA-sequencing results and to add some more molecular evidence to support their conclusion. Finally, since similar sex peptide signaling is not present in mammalians or humans, its implication in humans remains unclear.

---

## [Author Response]

The following is the authors’ response to the original reviews.

**Reviewer #1 (Recommendations For The Authors):**
My comments are largely limited to suggestions to make the manuscript easier to read and digest.In the abstract they say RNA sequencing highlights changes in innate...Could they be more specific? Innate immune system up or down? They do not indicate actual findings in the abstract.

We thank the reviewer for the comment and we have revised the abstract accordingly.

Their use of non‐intuitive abbreviations is often confusing. Perhaps they can add a table in methods listing all the abbreviations so that the reader can follow the data better. mNGA, vmHT....etc.

As suggested, we have now included a list of the abbreviations used in the paper.

There are mis‐spellings in the manuscript.

We have gone through the manuscript and corrected the mis-spellings.

Has the SPR RNAi line been validated?

The SPR RNAi line that we used has been extensively validated by Yapici et al., 2007 and several subsequent publications. Importantly, the effectiveness of SPR knockdown is evident in female flies as they exhibit dramatically reduced egg laying and, importantly, lack the typical post-mating behaviors (such as rejection of male flies after initial mating) observed in the wild type mated female flies. In fact, female flies with RNAi-mediated SPR knockdown behave identically to females mated with SP-null male flies, confirming the effective disruption of the SP-SPR signaling pathway. We have revised the manuscript and added these statements in the results section concerning SPR RNAi.

In the figures showing the Climbing Index vs time, can they abbreviate seconds as sec vs s? At least I think it is seconds. At first, I thought it was Time or Times, and was confused about what they were indicating on those types of graphs (Figures 1D‐F).

We have revised the figure as suggested by the reviewer.

In Figure 3F, they have a significance indicated in an unclear manner. It looks like they are comparing neuropil to the cortex, but I think they really mean to compare the cortex of sham to cortex of D31?

The reviewer was correct. We have revised figure 3F to make this clear.

In Figure 4B, what is the y‐axis? Percentage of what? Is that percentage of total flies?

The reviewer was correct. We have revised the figure to make this clear.

In a figure like SF3 B, what is the y‐axis? "Norm. Accum. CI" Can they explain the abbreviation?

We have revised the Y-axis label to be “Normalized accumulative CI”. We have also made this clear in the legend.

In the methods, what does this mean: "Regions devoid of Hoechst and phalloidin signal in non‐physiologically appropriate areas were considered vacuoles"? What are non‐physiologically appropriate areas? To me, that would mean outside of the brain. I would have thought the areas should be physiologically appropriate (aka neuropil and cortex)? This is confusing.

We have revised the method section to be more specific. In the *Drosophila* brain, there are structures such as esophagus that are devoid of both Hoechst and phalloidin staining, which were excluded from our vacuole quantification.

**Reviewer #2 (Recommendations For The Authors):**
Since I use mammalian systems, my comment about the confirmation of siRNA should be removed if this is not possible in the *Drosophila* system.

We have revised the figures to include total N values when appropriate. Including individual n values for each experimental assay and condition will inevitably crowd the figure legends, so specific values are available upon request.

Regarding RNAi knockdown of sex peptide receptors (SPRs), we agree that confirmation of the knockdown by IHC or qRT-PCR will further strengthen our findings. It should be noted, however, that the RNAi line we used has been extensively validated by Yapici et al., 2007 and several subsequent publications. Importantly, the effectiveness of SPR knockdown is evident in female flies as they exhibit dramatically reduced egg laying and, importantly, lack the typical post-mating behaviors (such as rejection of male flies after initial mating) observed in the wild type mated female flies. In fact, female flies with RNAi-mediated SPR knockdown behave identically to females mated with SP-null male flies, confirming the effective disruption of the SP-SPR signaling pathway. We have revised the manuscript to include these statements in the results concerning the SPR RNAi knockdown.

**Reviewer #3 (Recommendations For The Authors):**
(1) In Figures 1 and 2, the authors found that females have a lower climbing index in the acute phase in D17 injury, not due to neurodegeneration as shown no significant changes of brain vacuolation and other markers. However, in Figure 3, the authors found that female flies have a lower climbing index, more brain vacuolation, and neurodegeneration in the late phase. It's not very convincing that having a lower climbing index at the late phase is due to neurodegeneration. Is it possible that females suffered from more severe acute effects, at least in D17 injury?

We thank the reviewer for this point. Female flies injured on D17 displayed acute climbing deficits at 90 minutes post-injury. Since we did not observe significant structural changes in the brain at this time, we believe that this short-term functional deficit is not due to acute neuronal death. Here it is important to note that males did not display any acute climbing deficits when injured on D17, which suggests that the females suffered from more severe acute effects than males. However, these injured female flies recovered fully at 24 hours post-injury and displayed no climbing deficits. At two weeks post-injury, we observe climbing deficits and increased vacuole formation as a direct result of the injuries on D17 (see Supplemental Figure 3). When we assessed sensorimotor behavior and brain vacuolation on D45, we found that the injured females had significantly lower climbing indices and more brain vacuolation than the non-injured females of the same age. In this case, the concurrent observance of decreased climbing ability and increased brain vacuolation suggests chronic neurodegeneration in aged, injured females. This is not to be confused with the acute neuronal death observed by other groups using injury models of stronger severity. Overall, our data are consistent with the current view that in many neurodegenerative diseases, functional deficits often precede observable brain degeneration, which may take years to manifest.

(2) The authors determined late‐life brain deficits and neurodegeneration purely based on climbing index and vacuole formation. These phenotypes are not really specific to TBI‐related neurodegeneration and the significance and mechanisms of vacuole formation are not clear. Indeed, in Figures 3 A and B, male flies especially D31inj tend to have a much larger variation than any other groups. What could be the reasons? The authors should perform additional analyses on TBI‐related neurodegeneration in flies, which have been shown before, such as retinal degeneration and loss, neuronal degeneration, and loss, neuromuscular junction abnormalities, etc (Genetics. 2015 Oct; 201(2): 377‐402).

We thank the reviewer for the thorough evaluation of our manuscript. The reviewer raised a very important question: whether the neurodegeneration observed in our model is specific to TBI. As the reviewer rightly pointed out, the neurodegenerative phenotypes are unlikely to be specific to TBI-related neurodegeneration. Throughout the manuscript, we have tried to convey the notion that the mild physical impacts to the head represent one form of environmental insults, which in combination with other risk factors such as aging can lead to the emergence of neurodegenerative conditions. It should be noted that the negative geotaxis assay and vacuolation quantification are two well-established approaches to assess sensorimotor deficits and frank brain degeneration in fly brains.

It is important to emphasize that the head-specific impacts delivered to the flies in our study are much milder than those used in previous studies. As we showed in our figure 1, this very mild form of head trauma (referred to as vmHT) did not cause any death, nor affected the lifespan of the injured flies. Our supplemental data also show very minimal structural neuronal damage and no acute and chronic apoptosis induced by vmHT exposure. Consistently, we did not observe any exoskeletal or eye damage immediately following injuries, nor did we observe any retinal degeneration and pseudopupil loss at the chronic stage of these flies. We have incorporated these important points in the revised manuscript.

(3) In Figure 4, it would be important to perform the behavior test fly speed and directional movement in the acute phase as well to determine whether the females have reduced performance at the acute phase.

We thank the reviewer for this suggestion. Please note that our modified NGA has already improved the spatiotemporal resolution over the classic NGA. The data presented in Fig.3 show that there are no acute deficits for young cohorts. Therefore, we do not believe that the detailed analysis of the direction and speed of these flies is essential.

Unfortunately, the current setup for the AI-based analysis requires manual corrections of tracking errors, which are time-consuming and tedious. We are building a newly designed AI-based NGA (NGA.ai) that will allow automatic tracking and quantification with minimal manual interventions. Once it is completed, we will perform some of the analyses that the reviewer suggested.

(4) In Figure 8, the authors performed an RNA‐seq analysis and identified some dysregulated gene expressions. However, it is really surprising to see so few DEGs even in wild‐type males and mated females, and to see that none of DEGs overlap among groups or related to the SP‐signaling. This raises questions about the validity of the RNAseq analysis. It is critical to independently verify their RNA‐sequencing results and to add some more molecular evidence to support their conclusion.

We agree that future studies are needed to independently validate our RNA sequencing results. We believe that the small number of DEGs are likely due to two unique features of our study: (1) the very mild nature of our injury paradigm and (2) the chronic examination timepoint that was long after the head injury and SP exposure, which distinguish our study from previous fly TBI studies. As pointed out in the manuscript, our study was aimed to understand how early life exposure to repetitive head traumatic insults could lead to the latelife onset of neurodegenerative conditions. We hope to further validate our results in our next phase of experiments using single-cell RNA sequencing and RT-qPCR.

(5) The current results raise a series of interesting questions: what implication of female fly mating and its associated Sex Peptide signaling would be to mammalians or humans? Would mammalian female animals mating with wild‐type or sex hormone‐null male animals have different effects on their post‐injury behavior tests or neuropathological changes? What are the mechanisms underlying the sexual dimorphism?

As the reviewer pointed out, it would be very interesting to explore the possible roles of sex peptide-signaling in other animals and humans. As far as we know, there is no known mammalian ortholog to the insect sex peptide, so it would be difficult to study SP or an SPlike molecule in mammalian models. However, we believe that prolonged post-mating changes associated with reproduction in female fruit flies contribute to their elevated vulnerability to neurodegeneration. In this regard, drastic changes within the biology of female mammals associated with reproduction can potentially lead to vulnerability to neurodegeneration. We agree that this demands further study, which may be done with future collaborators using rodent or large animal models. We have discussed this point in the manuscript.